# Rapid Scoping Review: Empathy in Health Sciences Curriculum

**DOI:** 10.3390/healthcare11101429

**Published:** 2023-05-15

**Authors:** Renee Robinson, Kelleen Meluski, Tracy Hellem, Travis Hedwig, Natalie Hansen, Jennifer Adams, Mary Nies, Krista Salazar

**Affiliations:** 1College of Pharmacy, Idaho State University, Anchorage, AK 99508, USA; 2College of Pharmacy, University of New Mexico, Albuquerque, NM 87131, USA; 3Mark and Robyn Jones College of Nursing, Montana State University, Missoula, MT 59717, USA; 4College of Health, University of Alaska, Anchorage, AK 99508, USA; 5College of Pharmacy, Idaho State University, Meridian, ID 83642, USA; 6College of Health, Idaho State University, Pocatello, ID 83209, USA

**Keywords:** empathy, healthcare, higher education, curriculum, student, interprofessional

## Abstract

Empathy, the process of coming to know, understand, and care for another person, is a skill that can be learned; however, there is not a shared definition of empathy or understanding of how to operationalize empathy into practice. Healthcare worker empathy has been shown to have a beneficial effect on both patient health outcomes and the emotional wellness of healthcare workers. Empathic care is associated with more efficient, cost-effective, improved healthcare, and increased provider trust. The purpose of this rapid scoping review is to identify, compare, and contrast empathy training offered to select healthcare professional students (e.g., nurses, nurse practitioners, and pharmacists) as part of the general curriculum or as an elective. We utilized a rapid scoping review approach to identify potentially relevant peer-reviewed articles and studies for inclusion. Six electronic databases were searched, including: MEDLINE; EMBASE; PUBMed; CINAHL; EBSCOHOST; and ERIC, covering the past 10 years. A total of 4977 citations, 3480 abstracts, and 428 papers were screened. Fifty studies fulfilled the eligibility criteria. Of those, 21 primarily non-randomized experimental studies published between 2012 and 2021 were included in the final review. Over 80 percent of the training took place in the classroom setting and utilized active learning strategies. There is little consensus on how to best train future healthcare providers to provide empathic care to patients.

## 1. Introduction

Empathy, the process of coming to know, understand, and care for another person, is a skill that can be learned and developed over time [1,2]. Empathy involves not only feeling emotions but recognizing the emotions and putting yourself “in the shoes” of others [3]. Many individuals struggle to recognize/understand, share, and respond to social empathy cues, and this is particularly concerning in healthcare settings [4,5,6,7,8].

Healthcare worker empathy has been shown to have a beneficial effect on patient satisfaction and health outcomes, as well as the emotional wellness of healthcare workers [9,10]. The use of empathy in healthcare has been shown to have numerous benefits, including patient outcomes and increased provider trust. Additionally, empathy can help prevent medical paternalism, which continues to be a problem in the US healthcare system. A teachable and sustainable human resource, healthcare worker empathy is believed to be a necessary competency for all healthcare workers caring directly for patients (e.g., physicians, nurses, pharmacists, public health providers, etc.) and those working on interdisciplinary healthcare teams [4,11,12,13,14].

Empathic healthcare workers let patients know they are not alone and support patient self-efficacy, treatment adherence, and disease management planning, thus improving patient health outcomes and helping patients reach their health goals [15,16]. Empathic care is associated with more efficient, cost-effective healthcare (improved diagnostic accuracy) and increased provider trust [15,16,17]. Although no consensus definition of clinical empathy exists, empathy among current and future healthcare providers has been measured and the impact of both cognitive and affective elements of relationship-centered communication, a proxy for empathy, has been studied [1,18,19,20,21,22,23]. Studies suggest that healthcare worker empathy is associated with lower malpractice claims rates, job satisfaction, and reduced burnout [17].

Empathy training has been found to not only promote empathy but to prevent empathy deficits and erosion, thereby improving the ability of healthcare workers to counteract both prejudgments and stereotypes [24,25]. However, empathy training provided to healthcare workers does not appear to be standardized—it differs across healthcare worker/student groups and may erode over time, requiring additional training and support [2,20,21,22,23,26]. The purpose of this rapid scoping review is to identify, compare, and contrast empathy training offered to select healthcare professional students (e.g., nursing, pharmacy) as part of the general academic curriculum or as an elective.

## 2. Materials and Methods

We utilized a rapid scoping review approach in line with the methods recommended by Tricco in *BMC Medicine* and Arksey in *Int J Soc Res Methodology* to establish the scoping review protocol [27,28,29,30,31]. The protocol, search criteria, MESH terms, and inclusion criteria were revised based on feedback from a multidisciplinary research team and our medical librarian partners. Initial search terms included (empathy OR compassion OR sympathy OR humanism OR caring OR kindness) AND (curriculum OR curricula OR lesson OR program) AND (“health care”) AND (“higher education” OR school OR college OR graduate). For the search conducted in ERIC, the assessment included (empathy OR compassion OR sympathy OR humanism OR caring OR kindness) AND (curriculum OR curricula OR lesson OR program) AND (“health care”) AND (“higher education” OR school OR college OR graduate) AND “student”.

To identify potentially relevant peer-reviewed articles and studies for inclusion, six electronic databases were searched: MEDLINE; EMBASE; PUBMed; CINAHL; EBSCOHOST; and ERIC. The search was limited to the last 10 years (2012–2022) and was supplemented by a targeted literature search of relevant education journals in the identified three healthcare disciplines of interest (i.e., nurses, nurse practitioners, pharmacists). To ensure that all relevant studies were included, we reviewed the top 25 percent of articles that resulted from a Google Scholar search using the following search strategy: (empathy OR compassion OR sympathy OR humanism OR caring OR kindness) AND (curriculum OR curricula OR lesson OR program) AND (“health care”) AND (“higher education” OR school OR college OR graduate), which resulted in 3000 articles, of which the top 500 matches were reviewed. Lastly, a list of manuscripts that should be included in a “successful search” were established by the research team to confirm the appropriateness of search terms and plan.

Inclusion screening criteria were established a priori and focused on (1) empathy, the process of coming to know, understand, and care for another person [2], including the cognitive and affective elements related to communication or relationship building; (2) the population of interest (select healthcare professional students in one of the nursing and pharmacy and/or select medical providers (physician assistants, nurse practitioners); and (3) education, training, or other interventions offered to healthcare students as part of the required curriculum or as an elective, mandatory or optional lectures, and fully integrated, longitudinal, brief, or episodic curricular themes [2,16,23]. In addition, training studies outside the United States and studies that focused on post-graduate training were flagged for further review to determine if their empathy training study should be included in the scoping review.

All 3480 abstracts and 428 papers uploaded into covidence.org (an online, system review production tool for title/abstract screening, full-text screening, data abstraction, and quality assessment). While the studies varied in methodological quality and study design, they generally utilized active learning strategies in a classroom setting. This suggests that these strategies may be effective in promoting the development of empathic skills and attitudes among providers. All abstracts and full-text papers were independently reviewed by team-member pairs and all discrepancies resolved through research team discussion via Zoom (a secure, video conferencing platform). Note that all screening was completed independently using the agreed-upon data abstraction criteria within covidence.org.

## 3. Results

A total of 4977 citations and 428 potentially relevant full-text papers were screened for inclusion in the rapid scoping review. Fifty studies fulfilled the eligibility criteria and were included in the rapid review (Figure 1: PRISMA). A total of 20 studies, primarily non-randomized experimental studies published between 2012 and 2021, were included in the final review (Table 1 Summary Characteristics (Comprehensive Table Revised)). Training program type/intervention (e.g., lecture, simulation, theatrical performance), program length (1 h to 15 weeks in duration), empathy assessment tools employed (e.g., Pre–post Jefferson Scale of Empathy, Jefferson Scale of Physician Empathy—Health Professionals Version, Empathetic Communication Skills Scale, Empathetic Tendency Scale, Kiersma–Chen Empathy Scale, etc.) and study design (i.e., randomized controlled trial, non-randomized quasi-experimental design, pre–post survey, and qualitative assessment) varied significantly across training programs and disciplines. Several standardized self-assessment tools (e.g., Kiersma–Chen Empathy Scale and Jefferson Scale of Empathy) and program-developed empathy assessments were used to assess student empathy pre- and post-training participation. 

Over 80 percent of the training took place in the classroom setting and utilized active learning strategies (e.g., case-based assessments and discussions, self-reflection activities, peer-assisted learning techniques, etc.). Less than 10% of the studies provided the training program detail necessary to replicate, utilize, and/or test the training program efficacy and impact on healthcare student empathy, irrespective of site and/or discipline. Models and theoretical frameworks for learning (e.g., behaviorism, cognitive, constructivism, etc.) were only referenced in 10% of the identified studies; however, most training interventions focused on cognitive and/or cognitive and behaviorism frameworks. Models and theoretical frameworks for communication (e.g., relationship-centered communication, motivational interviewing, self-efficacy, etc.) were explicitly described in less than 10% of studies; however, close to 90% of the training interventions provided information about the education intervention used to teach students how to better communicate with patients (e.g., role-play, simulations, group discussions, etc.). Simulation laboratories and community settings were used in 20 percent of the identified training to mimic clinical practice and patient–provider interactions.

Pharmacists and nurses were equally represented in the selected studies, with one selected trial including both pharmacy and nursing students (Table 2: Empathy Training in Pharmacy versus Nursing Students). Active learning strategies (e.g., case-based discussions, simulations, and self-reflection, etc.) of similar duration (1 h to 1 semester in duration) were used by both pharmacy and nursing programs; however, only one program within the pharmacy curriculum measured students for multiple years in the program. Both pharmacy and nursing programs utilized standardized and program-developed assessment metrics to assess student empathy, employing student self-assessment and trainer (professor or clinician) assessment of student empathy. 

Seventy percent of the included empathy training program assessments were conducted in the United States (U.S.); however, over 50% of the nursing program assessments were conducted outside of the US (e.g., Palestine, Iran, Turkey, Australia, and Spain) (Table 3: empathy training in US versus international nursing students). International study designs were similar to those conducted in the US (e.g., pre- and post-test design, etc.) and variable empathy education duration/dose (e.g., one-time simulation, empathy education integrated in curricula delivered over the course of four years).

## 4. Discussion

Our scoping review highlights the current state of empathy training in healthcare professional education and the need for more standardized approaches to training healthcare professionals in empathic care. One of the benefits of using empathy in healthcare is that it can help prevent medical paternalism, which is a problem in the healthcare system. Without appropriate training and opportunities to practice, healthcare worker behaviors can make patients feel devalued and contribute to negative emotional states, aggression, and violence toward caregivers and other patients [16,49,50]. We believe that addressing this issue through empathy training can lead to improved patient outcomes and increased provider trust. 

Empathy training has been shown to reduce burnout and improve the ability of healthcare providers to care for challenging and often stigmatized populations (e.g., people who experience mental illness and those struggling with substance use disorders (MH/SUD)) [50,51,52,53,54]. However, there is little consensus on the most effective ways to train healthcare workers to provide empathic care and a wide range of instructional approaches are currently in use. Though a number of empathy training tools for healthcare students exist, it was difficult to assess the programs based on the information provided, including which tool was the most appropriate for students in specific healthcare disciplines, if the training addressed the unique needs of students providing care to individuals with mental health needs, and if the results were sustainable. There is an ongoing need to assess which educational strategies are most effective for which provider types and in which patient populations. This includes understanding which communication strategies have been most effective and best received by students and patients and assessing whether the effects of training erode over time or vary based on factors such as geography, community size, culture, or socioeconomic status.

### 4.1. Training Differences

In the scoping review, the authors found it difficult to assess the training programs based on the details provided. Empathy training varied significantly, from watching a one-hour video or participating in a brief three-dimensional simulation to extensive training structured into the curricula (semester-long training) [45]. Less than 10% of the studies provided the training program details required to replicate, utilize, and/or test the training program efficacy and impact of the training on healthcare student empathy, irrespective of site and/or discipline [45,47]. Models and theoretical frameworks for learning (e.g., behaviorism, cognitive, constructivism, etc.) were only referenced in 10% of the identified studies; however, the majority of training interventions focused on cognitive and/or cognitive and behaviorism [43]. In less than 10% of studies, models and theoretical frameworks for communication (e.g., relationship-centered communication, motivational interviewing, self-efficacy, etc.) were explicitly described; however, close to 90% of the training interventions provided information about the education intervention used to teach students how to better communicate with patients (e.g., role-play, simulations, group discussions, etc.), which allowed for some training program extrapolation [13,14,32,33,34,35,36,37,38,39,40,41,42,44,45,46,47,48].

Adults, in particular healthcare providers, can be taught to identify and empathically respond to a range of patient emotions (e.g., cognitive, emotional/behavioral, etc.) [4,5,7]. Cognitive empathy, understanding what another person might be feeling and thinking, allows the individual to take on their perspective in a situation. Emotional empathy, often referred to as clinical empathy, is the ability to share another person’s feelings, and focuses on understanding the patient’s feelings and using their experiences to skillfully adjust [9,10,23]. Though extrapolated in most of the identified studies, it appears that cognitive training methodologies, those most often employed in academia, were most commonly used to train pharmacy and nursing students in empathy, followed closely by emotional/behavioral frameworks [13,14,32,33,34,35,36,37,38,39,40,41,42,44,45,46,47,48]. Utilization of emotional/behavioral learning frameworks in empathy training program development may allow for improved understanding and subsequent adjustment of clinical care to better meet the needs of patients.

### 4.2. Differences between Training Based on Healthcare Student Type

Empathy is valued as a crucial component across all types of workplaces, from academic to corporate to the healthcare arena. Duration/intensity and training type varied significantly across nursing and pharmacy curricula, making it difficult to compare and assess the effectiveness of programs [13,13,14,16,32,33,34,35,36,38,39,41,42,43,45,46,48,55].

### 4.3. Differences between Training across Countries

Empathy education is delivered to health profession students occurs across the globe, although much of the published empathy education research was conducted in the United States. In this scoping review, six studies were conducted internationally, including in Palestine [32], Turkey [13,44], Australia [40], Spain [42], and Iran [43]. Simulation interventions evaluated were identified in countries across the globe [32,40], along with lectures [13,44] and workshops including role-playing, discussions, lectures, and scenarios [42,43]. There were no notable differences in the types of empathy interventions delivered internationally.

Again, the data collection tools used were identified in countries across the globe, e.g., the Kiersma–Chen Empathy Scale [32], Modified Kiersma–Chen Empathy Scale [40], Jefferson Scale of Empathy [42], and Jefferson Scale of Empathy for Health Professions Students [43]. Studies conducted in Turkey used tools that were unique to Turkey, such as the Empathic Communication Skill Scale and Empathic Tendency Scale [13,44].

Countries across the globe implemented similar study designs. For example, it was common to implement a pre–post-test design, and empathy education dose could be as short as a one-time simulation [40] as compared to empathy education integrated in curricula delivered over the course of four years [44]. All studies delivered education to nursing students and one study included nursing and medical students [13]. Findings were consistent across studies that empathy levels increased post-intervention; however, empathy appeared to erode over four years in the study that examined empathy education integrated in curricula [44].

One limitation of this scoping review is the variability in methodological quality and study design among the included studies. While efforts were made to assess the quality of each study using standardized tools, the use of primarily non-randomized experimental designs and the absence of consistent outcome measures make it difficult to draw firm conclusions about the effectiveness of different empathy training methods. In addition, while the review identified a trend towards utilizing active learning strategies in a classroom setting, there was little consensus on the specific training methods used across studies. Therefore, further research is needed to better understand the most effective approaches to training healthcare professional studies in empathic care and to develop standardized and validated methods for evaluating the effectiveness of empathy training.

## 5. Conclusions

Empathy training has been shown to improve the ability of healthcare providers to care for challenging and often stigmatized patients. However, there is little consensus on how to best train future healthcare providers to provide empathic care to patients and a number of questions remain. Models and theoretical frameworks for learning and communication (e.g., cognitive, behavioral/emotional, relationship-centered communication) and standardized assessment measures (e.g., Pre–post Jefferson Scale of Empathy, Jefferson Scale of Physician Empathy—Health Professionals Version, Empathetic Communication Skills Scale, Empathetic Tendency Scale, Kiersma–Chen Empathy Scale, etc.) need to be utilized in the development and testing of standardized empathy training programs across different healthcare students groups to better prepare our future healthcare providers.

## Figures and Tables

**Figure 1 healthcare-11-01429-f001:**
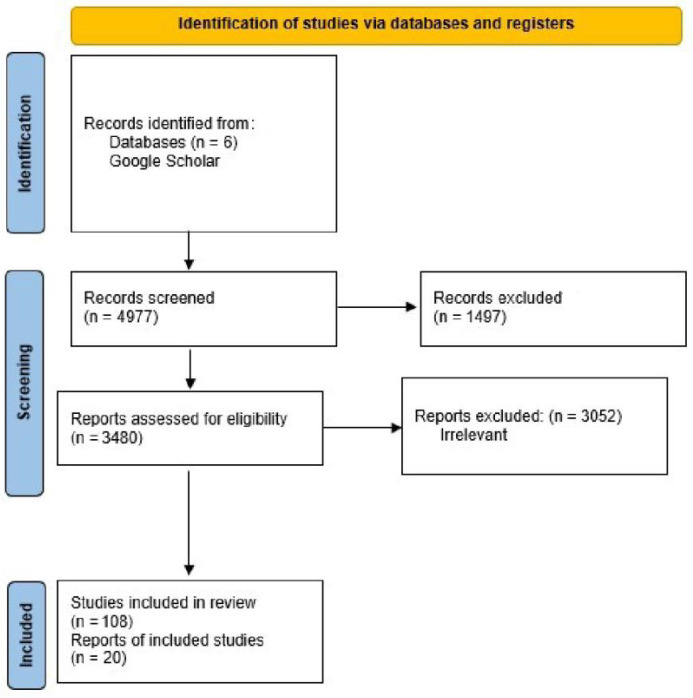
PRISMA flow diagram illustrating the process of study selection for the systematic literature review.

**Table 1 healthcare-11-01429-t001:** Comprehensive results table.

First Author (Last, First Middle Initial)	Year Published	Study Design	Population	Number of Participants	Country of Origin	Duration of Training (Number of Hours)	Educational Methodologies Implemented	Assessment Metric	Intervention Site	Outcome Metrics—Assessment Scale
Ayed, A. [32]	2021	Paired t-test was used to compare the differences between study variables pre–post-HFS intervention	Nursing	83	Palestine	<1 day per week for 3 weeks	Simulation-based learning, case-based learning	KCES, revised self-consciousness scale, patient–practitioner orientation scale, demographic data questionnaire	Simulation lab	Kiersma–Chen Empathy Scale
Chen, A.M.H. [33]	2015	Pre/post evaluation of an aging simulation game and its impact on pharmacy students’ empathy	Pharmacy	156	United States	3 h	Case-based learning, simulations, gaming, reflection, discussion	Kiersma–Chen Empathy Scale and Jefferson Scale of Empathy	Learning laboratory	Kiersma–Chen Empathy Scale and Jefferson Scale of Empathy
Adams, S. [34]	2019	Cross-sectional descriptive methods	Nursing	101	United States	3 h			Classroom	Oral debriefing followed by written reflections
Ozcan, C.T. [13]	2012	Non-randomized experimental study (pre-/post quasi-experimental design)	Nursing and medical students	226	Turkey	10 h total (2 h per week for 5 weeks)	None stated	Empathic Communication Skills Scale (ECSS) and the Empathic Tendency Scale (ETS)	Simulation lab	Empathic Communication Skills Scale (ECSS) and the Empathic Tendency Scale (ETS)
Miller, S.R. [35]	2013	Qualitative research	Nursing; Pharmacy; Physician Assistants	95	United States	1–3 h	Observational learning, self-reflection	ATDP, AMIA	Classroom	Attitude toward Disabled Persons scale (ATDP), the Attitudes toward patient advocacy Microsocial scale (AMIA), and the Interpersonal Reactivity Index
Ekong, G. [36]	2017	Cross-sectional study	Pharmacy	134	United States	<1 day, MI development—3 weeks of skilled lab sessions	Active Learning Activities, discussions, case-based assessment	OSCE, KCES, ISS, Self-report	Classroom	KCES and ISS
Underwood, M. [37]	2020	Cross-sectional study	Pharmacy	50	United States	7-day immersive experiential	Case-based learning	KCES	Classroom	Kiersma–Chen Empathy Scale
Zelenski, A.B. [14]	2020	Non-randomized experimental study	Nursing; Pharmacy; Physician Assistants; Public Health	86	United States	15 h (6 times over 6–8 weeks for 2.5 h per session)	Improv is about attunement, affirmation, and advancement	IRI, CARE (Consultative and Relational Empathy)	Classroom	Interpersonal Reactivity Index [IRI], Consultative and Relational Empathy [CARE] measure
Van Winkle, L.J. [38]	2012	Non-randomized experimental study	Pharmacy	233	United States	40 min	Self-reflection, case-based learning, simulation-based learning, discussion	JSE pre and post		JSE (HPS-Version for pharmacy students and JSE, S-Version, for medical students
Lor, K.B. [39]	2015	Randomized controlled trial	Pharmacy	20 intervention, 20 control	United States	3-h	Simulation-based learning, peer-assisted learning, observational learning	Pre- and post-Jefferson Scale of Empathy—Health Profession Students version (JSE-HPS)	Touro University, College of Pharmacy, Vallejo California; Keck Graduate Institute School of Pharmacy, Claremont, CA	Jefferson Scale of Empathy—Health Profession Students version (JSE-HPS)
Everson, N. [40]	2015	One-group pretest, post-test design	Nursing	460 s year students	Australia	<1 h	3D simulation viewing, debrief, reflection	Pre- and post-Modified Kiersma–Chen Empathy Scale, graded reflection	The University of Newcastle, School of Nursing, Australia	Modified Kiersma–Chen Empathy Scale
Isaacs, D. [41]	2015	Non-randomized experimental study	Pharmacy	130 students	United States	15 weeks	Chronic disease simulation, chronic disease state reflection questions	Self-report survey, writing reflection, faculty-developed rubric (SOAP), ACPE standards linked to simulation	Chicago State University College of Pharmacy, Chicago, Illinois	Self-report survey, writing reflection, faculty-developed rubric (SOAP), ACPE standards linked to simulation
Bas-Sarmiento, P. [42]	2017	Quasi-experimental study of a single group with pre-test–post-test measurements	Nursing	48 (second year nursing students)	Spain	<1 day	Simulation-based learning, role-playing, problem-based learning, simulated clinical interviews	Independent external observer, Jefferson Scale of Empathy (JSE), Reynolds Empathy Scale (RES), Consultation and Relational Empathy (CARE) Measure	University of Cadiz, School of Nursing, SW Spain	Jefferson Scale of Empathy, Reynolds Empathy Scale, Consultation and Relational Empathy Measure
Gholamzadeh, S. [43]	2018	Randomized controlled trial	Nursing	63 (32 intervention group) 3rd and 4th year students	Iran	<1 day (8 h)	Lecture, case-based learning, role-play and simulations, exercise on self-awareness and empathy skills, listening skills	Kogan’s Attitudes towards old people scale (KAOP), Jefferson Scale of Physician Empathy—Health Professionals Version (JSPE-HP)	Hazrat Zeinab Nursing and Midwifery School of Yazuj, Iran	Jefferson Scale of Physician Empathy—Health Professionals Version
Ozcan, C.T. [44]	2018	Non-randomized experimental study	Nursing	66	Turkey	Semester curricula	Case-based learning, simulation-based learning, peer-assisted learning, discussion	Reflections, discussion	State university school of nursing in Ankara, Turkey	Empathic Communication Skills Scale (ECSS) and the Empathic Tendency Scale (ETS)
Parker, D. [45]	2019	Non-randomized experimental study (pre/post-survey)	Pharmacy	50 P2 and P3 students	United States	3-day simulation part of 15 week elective—2 h/week	Hypothesized simulation-based learning activity, instructor-facilitated class discussion	Pre–post-survey, questions 1–6 designed to assess empathy, questions 7–16 designed to assess student self-confidence in diabetes management	University of Findlay, College of Pharmacy, Findlay, Ohio	Pre- and post-activity survey
Bailey-Preston, K. [46]	2020	Non-randomized experimental study (survey)	Pharmacy	35	United States	Pro Me—outreach performance, 2 per semester	Creative Volunteerism to promote educational wellbeing	Summative 5-point Likert scale survey with open-ended questions to assess various aspects of the experience including empathy level, perceived impact on creativity, and lessons learned	Appalachian College of Pharmacy, Oakwood, VA—performances at assisted living and long-term care facilities, hospitals, mobile medical clinics, health fairs, and schools	A brief survey was sent to active members of the organization and a 5-point Likert scale was used to assess various aspects of student experience, including improvements in empathy level
Nebergall, S. [47]	2021	Non-randomized experimental study (qualitative and quantitative data)	Pharmacy	119-P1; 152 P2, 105	United States	Patient voices Course (2 h lecture, 3 h skills lab) 4-P1, 2-P2, 2-P3 and 40-hour IPPE -P1	Lecture, patient speaker, lab activity—simulation-based learning (fishbowl style), peer-assisted learning	IPPE preceptor evaluation, minute paper reflections student self-ratings (Empathy and Cultural Sensitivity)	The Ohio State University College of Pharmacy	Pre–post-survey (2 questions to assess confidence in student ability to show empathy and confidence in their ability to counsel patients from marginalized communities, IPPE preceptor and student self-evaluations (4-point scale Entrustable Professional Activities—EPA survey), open-ended fields to identify most valuable components, minute paper
Clemens, E. [48]	2021	Pre/post-evaluation of course elective	Pharmacy	27	United States	12 weeks (hours not mentioned)	Didactic lectures, instructional videos, in-class discussions, patient speakers	Writing assignments (bucket list, self-obituary, reflections based on material), self-reflection, KCES pre- and post-course survey, fear of death and dying scale (CL-FODS), Frommelt Attitudes towards care of the dying (FATCOD-B) scale Form B Assessment	Pursue University, College of Pharmacy	Kiersma–Chen Empathy Scale (KCES), Collett–Lester Fear of Death and Dying Scale (CL-FODS), Frommelt Attitudes Toward Care of the Dying Scale Form B (FATCOD-B)

**Table 2 healthcare-11-01429-t002:** Empathy training in pharmacy versus nursing students.

First Author	Education Intervention (i.e., Role Play, Reflection, Lecture)	Duration	Assessment Metrics
Nursing			
Adams, S. [34]	Game	3 h	Oral debriefing followed by written reflections
Ayed, A. [32]	Simulation-based learning, case-based learning	<1 day per week for 3 weeks	KCES, revised self-consciousness scale, patient–practitioner orientation scale, demographic data questionnaire
Bas-Sarmiento, P. [42]	Simulation-based learning, role-playing, problem-based learning, simulated clinical interviews	<1 day	Jefferson Scale of Empathy, Reynolds Empathy Scale, Consultation and Relational Empathy Measure
Everson, N. [40]	3D simulation viewing, debrief, reflection	<1 h	Pre- and post-Modified Kiersma–Chen Empathy Scale, graded reflection
Gholamzadeh, S. [43]	Lecture, case-based learning, role-play and simulations, exercise on self-awareness and empathy skills, listening skills	<1 day (8 h)	Kogan’s Attitudes towards old people scale (KAOP), Jefferson Scale of Physician Empathy—Health Professional Version (JSPE-HP)
Miller, S.R. [35]	Observational learning, self-reflection	1–3 h	Attitude toward Disabled Persons scale (ATDP), the Attitudes toward Patient Advocacy Microsocial scale (AMIA), and the Interpersonal Reactivity Index
Ozcan, C.T. [13]	Oral debriefing followed by written reflections	10 h total (2 h per week for 5 weeks)	Empathic Communication Skills Scale (ECSS) and the Empathic Tendency Scale (ETS)
Ozcan, C.T. [44]	Case-based learning, simulation-based learning, peer-assisted learning, discussion	Semester curricula	Empathic Communication Skills Scale (ECSS) and the Empathic Tendency Scale (ETS)
Zelenski, A.B. [14]	Improv is about attunement, affirmation, and advancement	15 h (6 times over 6–8 weeks for 2.5 h per session)	Interpersonal Reactivity Index [IRI], Consultative and Relational Empathy [CARE] measure
Pharmacy			
Bailey-Preston, K. [46]	Creative Volunteerism to promote educational wellbeing	Pro Me—outreach performance, 2 per semester	Summative 5-point Likert scale survey with open-ended questions to assess various aspects of the experience including empathy level, perceived impact on creativity, and lessons learned
Chen, M.H. [33]	Case-based learning, simulations, gaming, reflection, discussion	3 h	Kiersma–Chen Empathy Scale and Jefferson Scale of Empathy
Clemens, E. [48]	Didactic lectures, instructional videos, in-class discussions, patient speakers	12 weeks (hours not mentioned)	Kiersma–Chen Empathy Scale (KCES), Collett-Lester Fear of Death and Dying Scale (CL-FODS), Frommelt Attitudes Toward Care of the Dying Scale Form B (FATCOD-B)
Ekong, G. [36]	Active Learning Activities, discussions, case-based assessment	<1 day, MI development—3 weeks of skilled lab sessions	OSCE, KCES, ISS, Self-report
Isaacs, D. [41]	Chronic disease simulation, chronic disease state reflection questions	15-weeks	Self-report survey, writing reflection, faculty-developed rubric (SOAP), ACPE standards linked to simulation
Lor, K.B. [39]	Simulation-based learning, peer-assisted learning, observational learning	3 h	Jefferson Scale of Empathy—Health Profession Students version (JSE-HPS)
Nebergall, S. [47]	Lecture, patient speaker, lab activity—simulation-based learning (fishbowl style), peer-assisted learning	Patient voices course (2 h lecture, 3 h skills lab) 4-P1, 2-P2, 2-P3, and 40 h IPPE -P1	Pre–post-survey (2 questions to assess confidence in student ability to show empathy, and confidence in their ability to counsel patients from marginalized communities), IPPE preceptor and student self-evaluations (4-point scale Entrustable Professional Activities—EPA survey), open-ended fields to identify most valuable component, minute paper
Parker, D. [45]	Hypothesized simulation-based learning activity, instructor-facilitated class discussion	3-day simulation part of 15-week elective—2 h/week	Pre–post-survey, questions 1–6 designed to assess empathy, questions 7–16 designed to assess student self-confidence in diabetes management
Underwood, M. [37]	Case-based learning	7-day immersive experiential	Kiersma–Chen Empathy Scale
Van Winkle, L.J. [38]	Self-reflection, case-based learning, simulation-based learning, discussion	40 min	JSE (HPS-Version) for pharmacy students and JSE (S-Version) for medical students

**Table 3 healthcare-11-01429-t003:** Nursing student empathy training identified in countries across the globe.

First Author	Country	Education Intervention	Duration	Outcome Metrics
Adams, S. [34]	United States	Game	3 h	Oral debriefing followed by written reflections
Ayed, A. [32]	Palestine	Simulation-based learning, case-based learning	<1 day per week for 3 weeks	KCES, revised self-consciousness scale, patient–practitioner orientation scale, demographic data questionnaire
Bas-Sarmiento, P. [42]	Spain	Simulation-based learning, role-playing, problem-based learning, simulated clinical interviews	<1 day	Jefferson Scale of Empathy, Reynolds Empathy Scale, Consultation and Relational Empathy Measure
Everson, N. [40]	Australia	3D simulation viewing, debrief, reflection	<1 h	Pre- and post-Modified Kiersma–Chen Empathy Scale, graded reflection
Gholamzadeh, S. [43]	Iran	Lecture, case-based learning, role-play and simulations, exercise on self-awareness and empathy skills, listening skills	<1 day (8 h)	Kogan’s Attitudes towards old people scale (KAOP), Jefferson Scale of Physician Empathy—Health Professional Version (JSPE-HP)
Miller, S.R. [35]	United States	Observational learning, self-reflection	1–3 h	Attitude toward Disabled Persons scale (ATDP), the Attitudes toward Patient Advocacy Microsocial scale (AMIA), and the Interpersonal Reactivity Index
Ozcan, C.T. [13]	Turkey	Oral debriefing followed by written reflections	10 h total (2 h per week for 5 weeks)	Empathic Communication Skills Scale (ECSS) and the Empathic Tendency Scale (ETS)
Ozcan, C.T. [44]	Turkey	Case-based learning, simulation-based learning, peer-assisted learning, discussion	Semester curricula	Empathic Communication Skills Scale (ECSS) and the Empathic Tendency Scale (ETS)
Zelenski, A.B. [14]	United States	Improv is about attunement, affirmation, and advancement	15 h (6 times over 6–8 weeks for 2.5 h per session)	Interpersonal Reactivity Index [IRI], Consultative and Relational Empathy [CARE] measure

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
