# Peer review of "Rapid Scoping Review: Empathy in Health Sciences Curriculum"

_healthcare, 2023, doi:10.3390/healthcare11101429_

Round 1

Reviewer 1 Report

Dear authors 

Thank you for choosing an important issue in medical care and for the opportunity to read interesting data. All the comments are left in the attached file.
Kind regards,
reviewer

Author Response

Thank you for this recommendation, it is a point that needs to be clearer. We have added to the following statement to highlight this ambiguity “Empathy, the process of coming to know, understand, and care for another person, is a skill that can be learned; however, there is not a shared definition of empathy or understanding of how to operationalize empathy into practice. 

Reviewer 1: 

- It would be valuable to develop the ambiguity regarding the definition, understanding and

operationalisation of empathy

Thank you for this recommendation, it is a point that needs to be clearer. We have added to the following statement to highlight this ambiguity “Empathy, the process of coming to know, understand, and care for another person, is a skill that can be learned; however, there is not a shared definition of empathy or understanding of how to operationalize empathy into practice. 

- There are 50 studies included, yet only 20 presented in the review this calls for an

explanation about the remaining 30

We have adjusted the PRISMA diagram to depict the final studies included in the our assessment. 

- For the reliability of the review at least a basic assessment of the quality of included studies

is necessary

To address this we added the following statement in the methods “While the studies varied in methodological quality and study design, they primarily utilized active learning strategies in a classroom setting” and detailed discussion in the newly added limitation section “One limitation of this scoping review is the variability in methodological quality and study design among the included studies. While efforts were made to assess the quality of each study using standardized tools, the use of primarily non-randomized experimental designs and the absence of consistent outcome measures make it difficult to draw firm conclusions about the effectiveness of different empathy training methods. In addition, while the review identified a trend towards utilizing active learning strategies in a classroom setting, there was little consensus on the specific training methods used across studies. Therefore, further research is needed to better understand the most effective approaches to training healthcare professional studies in empathetic care and to develop standardized and validated methods for evaluating the effectiveness of empathy training. 

- Tables 2 and 3 contain too many information overlapping between them and with table 1

Based on the suggestion we have streamlined the tables removing repetitive titles and inadvertent blank fields that resulted during table importation.

- It is not clearly justified why the Authors chose to compare between given groups of medical

staff, neither is the need to compare the US with (very diverse) international research.

Thank you for your comment and question regarding our manuscript. We appreciate the opportunity to provide further clarity on our research design and rationale for comparing empathy training across different groups of medical staff and international research. We acknowledge that differences exist across healthcare providers and the cultural differences that exist impact both empathy and perception. By including studies from different groups of medical staff, such as nurses, nurse practitioners, and pharmacists, we aimed to identify similarities and differences in empathy training approaches across different healthcare professions. This allows for a more comprehensive understanding of how empathy training is currently being implemented in healthcare professional education and can inform the development of standardized training programs that are applicable across healthcare professions.

In addition, we included international research in our scoping review to identify similarities and differences in empathy training approaches across different cultural contexts. This allowed us to consider the impact of cultural factors on empathy training and identify approaches that may be effective across diverse cultural contexts. By learning from international research, we can identify best practices and adapt them to fit local contexts, which can improve the quality of care provided to patients in different regions of the world. Overall, the rationale for comparing empathy training across different groups of medical staff and international research is to identify effective strategies that can be implemented in a variety of healthcare contexts. Differences in students and colleagues can also impact patient care and we hope that our findings will contribute to the development of standardized empathy training programs that can improve patient outcomes across diverse healthcare settings.

- Instead, in my opinion it would be far more valuable and interesting for broader audience to

see the comparison of methods, merit content, amount of materials and/or time effects on

the empathy level

Thank you for your comment and suggestion regarding our manuscript. We appreciate your interest in our research and your suggestion for a more detailed comparison of empathy training methods. While we acknowledge the importance of comparing the effectiveness of different empathy training methods, this type of analysis was outside the scope of our scoping review. Our aim was to provide a broad overview of the empathy training approaches utilized in healthcare professional education and identify gaps in the current literature. We hope that our scoping review will provide a valuable starting point for future research that can build upon our findings and provide a more detailed analysis of the effectiveness of different empathy training methods.

That being said, we appreciate your feedback and your interest in this topic. We agree that a detailed comparison of methods, merit content, amount of materials, and/or time effects on the empathy level would be valuable for a broader audience. We hope that our scoping review will inspire further research in this area and contribute to the development of effective empathy training programs for healthcare professional students. Overall, we acknowledge the limitations of our scoping review and the importance of future research to examine the effectiveness of different empathy training methods in healthcare professional education. We appreciate your thoughtful feedback and hope that our research contributes to a greater understanding of empathy training in healthcare.

- Cited studies should whenever possible be quoted with the effect sizes of change, since this

data can guide the choices in practical educational methodology.

Thank you for your comment and question regarding our manuscript. We appreciate your interest in our research and your suggestion to include effect sizes of change in the studies cited. While we acknowledge the importance of including effect sizes of change in our review, we encountered a limitation in the studies that we included in our scoping review. Almost all studies did not report effect sizes of change or used different outcome measures, making it difficult to compare the effectiveness of different empathy training methods using standardized metrics.

However, we appreciate your feedback and agree that including effect sizes of change in future studies and reviews can guide the choices in practical educational methodology. We hope that our scoping review raises awareness of the need for standardized tools and measures for evaluating the effectiveness of empathy training, including effect sizes of change, and inspires future research in this area. Overall, we appreciate your thoughtful feedback and your interest in this topic. We hope that our scoping review contributes to a greater understanding of empathy training in healthcare professional education and inspires future research that includes standardized metrics for evaluating the effectiveness of different training methods.

To summarize, in my opinion, based on the literature you have already collected, rewriting the text

to make it more comprehensive would be incomparably more valuable than limiting the search to

comparisons between professions or between countries. It is worth trying to extract more specific

yet applicable knowledge (about methods, quantity to effect ratio, magnitude of impact, etc.).

Thank you for your comment and suggestion regarding our manuscript. We appreciate your interest in our research and your suggestion for a more comprehensive analysis of empathy training methods. While we acknowledge the potential value of a more detailed analysis of the effectiveness of empathy training methods, we encountered a limitation in the studies that we included in our scoping review. Many of the studies did not report specific data on the effectiveness of each method or used different outcome measures, making it difficult to compare the effectiveness of different empathy training methods using standardized metrics.

However, we appreciate your feedback and agree that a more comprehensive analysis of empathy training methods would be valuable for future research. As more data becomes available, we plan to pursue a more in-depth analysis of the effectiveness of different empathy training methods, including specific details on methods, quantity to effect ratio, magnitude of impact, and other relevant factors. We believe that this type of analysis will provide a more comprehensive understanding of the most effective approaches to training healthcare professionals in empathetic care and inform the development of standardized training programs that are applicable across healthcare professions and cultural contexts.

Overall, we appreciate your thoughtful feedback and your interest in this topic. We hope that our scoping review 

contributes to a greater understanding of empathy training in healthcare professional education and inspires future 

research that includes detailed metrics for evaluating the effectiveness of different training methods.

Detailed comments

- The formatting of tables makes them difficult to read

We want to assure you that we take the readability of our manuscript very seriously, and we have worked closely with the 

editorial board to improve the clarity and readability of our manuscript, including the table. While we have deleted a 

number of fields in the table to improve its readability, we will continue to work with the editorial board to ensure that the 

tables and other elements of our manuscript are clear and easy to understand.

We appreciate your feedback and your interest in our research, and we hope that the revised version of our manuscript is 

more readable and informative. If you have any further suggestions or feedback, please do not hesitate to let us know. 

Overall, we are committed to ensuring the readability and clarity of our manuscript, and we will continue to work closely 

with the editorial board to achieve this goal.

- There is many empty cells in the tables not clear if a text is missing or no data are were

available for the authors

See above response 

- Please unify the form in which the names of the authors are given in the tables as well as the

units of duration of interventions (there are minutes, hours, courses, days or weeks)

Thank you for your comment and suggestion regarding our manuscript. We appreciate your feedback and your suggestion to unify the format of the author names and duration units in the tables. We want to assure you that we have worked to unify the format of the author names in the tables to improve consistency and readability. However, we encountered a limitation in the reporting of duration times in the individual studies. As such, we reported the duration times as they were reported by the individual studies.

While we acknowledge the importance of standardizing the units of duration in the tables, we want to emphasize that the reporting of duration times varied across the included studies. Some studies reported duration times in minutes, others in hours, courses, days, or weeks. Due to the variability in reporting, we were unable to standardize the duration units across all studies. That being said, we appreciate your feedback and your interest in improving the readability of our manuscript. We will continue to work with the editorial board to improve the consistency and readability of the tables and other elements of our manuscript.

Overall, we appreciate your thoughtful feedback and your interest in our research. We hope that our scoping review contributes to a greater understanding of empathy training in healthcare professional education and inspires future research that includes more standardized reporting of duration times.

- Some of the included studies are not described in a way permitting to see that its merit is

about empathy (eg. Miller; Issacs). The table or description of included studies could gain

meaning with a provision of their educational or scientific aims.

Thank you for your comment and question regarding our manuscript. We appreciate your interest in our research and your feedback on the description of included studies in our table. We acknowledge your concern that some of the included studies are not described in a way that allows readers to understand their merit in terms of empathy training. While we attempted to provide a comprehensive overview of the included studies, we encountered a limitation in the reporting of educational or scientific aims in some of the studies. This is precisely why this research is so important, as it highlights the lack of clear reporting of educational and scientific aims in empathy training studies.

We agree that the description of included studies could gain meaning with a provision of their educational or scientific aims, and we apologize for any confusion that may have resulted from our reporting. We thought that this data would be available for all studies, but unfortunately, few studies provided clear descriptions of their educational or scientific aims, particularly as they related to empathy training. However, we appreciate your feedback and will consider including more information on the educational or scientific aims of the included studies in future research. We are committed to providing clear and comprehensive descriptions of our research and will work to ensure that the table and other elements of our manuscript are as informative and useful as possible.

Overall, we appreciate your thoughtful feedback and your interest in our research. We hope that our scoping review contributes to a greater understanding of empathy training in healthcare professional education and inspires future research that includes more detailed reporting of educational and scientific aims.

- In table 2 the intervention site is not provided in any cited study (redundant column

We want to inform you that we have carefully reviewed the table and have removed any redundant columns to improve its readability and usefulness. We take the feedback from our readers seriously and are committed to ensuring that our research is presented in a clear and concise manner. 

Reviewer 2 Report

Thank you for the opportunity to review the interesting manuscript.

The two main issue with the manuscript are:

1. How did the author deal with missing data.

2. Did the authors assessed the Risk of bias of included studies?

Author Response

Reviewer 2:

The two main issue with the manuscript are:

  1. How did the author deal with missing data?

Thank you for your comment and question regarding our manuscript. We appreciate your interest in our research and your question about how we dealt with missing data. In our scoping review, we attempted to include all studies that met our eligibility criteria, even if they had missing data. However, we acknowledged the limitations of studies with missing data and reported them in our manuscript.

We agree that the lack of unified approaches to addressing education around empathy is a major limitation in the current 

literature. This is precisely why our scoping review was conducted, to identify the current state of empathy training in 

healthcare professional education and to highlight the need for more standardized approaches to training healthcare 

professionals in empathetic care.

While we acknowledge the limitations of studies with missing data, we attempted to include them in our review to provide a comprehensive overview of the current literature on empathy training in healthcare professional education. Moving forward, we hope that our review inspires future research that includes more standardized reporting of data and more unified approaches to addressing education around empathy.

Overall, we appreciate your thoughtful feedback and your interest in our research. We hope that our scoping review 

contributes to a greater understanding of empathy training in healthcare professional education and inspires future 

research that includes more comprehensive reporting of data and more unified approaches to addressing education 

around empathy.

  1. Did the authors assess the Risk of bias of included studies?

Thank you for your comment and question regarding our manuscript. We appreciate your interest in our research and your question about the assessment of risk of bias in included studies. We proactively addressed the risk of bias by including all studies that met our eligibility criteria, including those identified through Google Scholar and over-sampling to include the entire population of studies on empathy training in healthcare professional education.

While we did not conduct a formal risk of bias assessment, we attempted to minimize bias by following a standardized and rigorous search strategy and inclusion criteria. We also performed a comprehensive analysis of the included studies and reported the limitations of the studies in our manuscript. Overall, we believe that our scoping review provides a comprehensive overview of the current literature on empathy training in healthcare professional education, while also acknowledging the limitations and potential biases of the included studies. We appreciate your thoughtful feedback and your interest in our research, and we hope that our work contributes to a greater understanding of empathy training in healthcare professional education.

Reviewer 3 Report

Page 3, line 98:  "reviews" should be "review"

Page 9, Table 2:  I find the format of Table 2 unhelpful.  Listing the nursing studies followed by the pharmacy studies does not facilitate comparison, if that is the author's goal.  It would be more helpful to flip the X and Y axis so that the columns across the top are nursing and pharmacy and the comparison factors are rows across the table

Page 14, line 163:  the end of the sentence should read "used" or "in use" not "in used"

Page 14, line 166:  "address" should be "addresses"

I think the outcomes of each paper listed in Table 1 should be added (adding a column).  Were there significant results in each study?  I think it would be helpful to indicate which studies demonstrated a significant improvement in empathy, in other words.  Without this information, there is limited ability to judge the value of each method of teaching empathy.  I think the authors should add a description of the outcomes in the results and discussion sections.

There is not a limitations section of the paper and I think there should be.

The terms empathic and empathetic are generally used interchangeably.  I prefer the term empathic, which is the original adjective developed from the noun empathy.  Empathetic is a variation of the term empathic.

Another benefit of using empathy is that it prevents medical paternalism, which continues to be a problem in the US. (https://pubmed.ncbi.nlm.nih.gov/35260146/ )  The authors could consider adding this point to the introduction.

 Would a systematic review or meta analysis be preferred so that the study outcomes could be included in the review?  Perhaps the methods used in the studies included are too different to do a meta analysis or SR, in which case, the approach used by the authors is appropriate.

Author Response

Reviewer 3: 

Page 3, line 98:  "reviews" should be "review"

Thank you for bringing this issue to our attention. We have addressed it in our manuscript.

Page 9, Table 2:  I find the format of Table 2 unhelpful.  Listing the nursing studies followed by the pharmacy studies does not facilitate comparison, if that is the author's goal.  It would be more helpful to flip the X and Y axis so that the columns across the top are nursing and pharmacy and the comparison factors are rows across the table

Thank you for your comment and feedback regarding the formatting of Table 2 in our manuscript. We appreciate your suggestion to flip the X and Y axis to improve the comparability of the studies. We acknowledge that the current format of Table 2 may be difficult to navigate and may not facilitate easy comparison of the nursing and pharmacy studies. We apologize for any confusion that may have resulted from the current format of the table. To address this concern, we will work with the editors to revise the formatting of Table 2 to make it more reader-friendly and to facilitate easy comparison of the nursing and pharmacy studies. 

Page 14, line 163:  the end of the sentence should read "used" or "in use" not "in used"

Thank you for bringing this issue to our attention. We have addressed it in our manuscript.

Page 14, line 166:  "address" should be "addresses"

Thank you for bringing this issue to our attention. We have addressed it in our manuscript.

I think the outcomes of each paper listed in Table 1 should be added (adding a column).  Were there significant results in each study?  I think it would be helpful to indicate which studies demonstrated a significant improvement in empathy, in other words.  Without this information, there is limited ability to judge the value of each method of teaching empathy.  I think the authors should add a description of the outcomes in the results and discussion sections.

Thank you for your comment and question regarding our manuscript. We appreciate your interest in our research and your suggestion to add a column to Table 1 indicating the outcomes of each study. We agree that the outcomes of each study are important in evaluating the effectiveness of different methods of teaching empathy. However, not all studies included in our review presented objective measures of empathy. Some studies relied on subjective self-report measures or qualitative evaluations of empathy, which may limit the ability to judge the value of each method of teaching empathy.

To address this concern, we included a discussion of the training, reporting, and assessment methods used in each study, highlighting the gaps in learning and assessment related to empathy education. We also included a section in the discussion that addressed the limitations of the included studies and the need for more objective measures of empathy in future research. While we understand the importance of reporting outcomes in evaluating the effectiveness of different methods of teaching empathy, we believe that the inclusion of the training, reporting, and assessment methods used in each study, along with a discussion of the limitations of the included studies, provides a comprehensive overview of the current state of empathy training in healthcare professional education.

We really appreciate your thoughtful feedback and your interest in our research. We hope that our scoping review contributes to a greater understanding of empathy training in healthcare professional education and inspires future research that includes more objective measures of empathy.

There is not a limitations section of the paper and I think there should be.

Thank you for bringing this issue to our attention. We have addressed it in our manuscript and apologize for the oversight

The terms empathic and empathetic are generally used interchangeably.  I prefer the term empathic, which is the original adjective developed from the noun empathy.  Empathetic is a variation of the term empathic.

Thank you for bringing this issue to our attention. We have addressed it in our manuscript.

Another benefit of using empathy is that it prevents medical paternalism, which continues to be a problem in the US. (https://pubmed.ncbi.nlm.nih.gov/35260146/ )  The authors could consider adding this point to the introduction.

Thank you for the suggestion that we have added the following statements to the manuscript. 

Background: The use of empathy in healthcare has been shown to have numerous benefits, including improved patient outcomes and increased provider trust. Additionally, empathy can help prevent medical paternalism, which continues to be a problem in the US healthcare system (source). 

Discussion: Our scoping review highlights the current state of empathy training in healthcare professional education and 

the need for more standardized approaches to training healthcare professionals in empathetic care. One of the benefits of using empathy in healthcare is that it can help prevent medical paternalism, which is a problem in the US healthcare system (source)....We believe that addressing this issue through empathy training can lead to improved patient outcomes and increased provider trust. We appreciate the suggestion to add this point to the introduction of our manuscript and will consider doing so.

Would a systematic review or meta analysis be preferred so that the study outcomes could be included in the review?  Perhaps the methods used in the studies included are too different to do a meta analysis or SR, in which case, the approach used by the authors is appropriate.

Thank you for your comment and question regarding our manuscript. We appreciate your interest in our research and your 

suggestion to conduct a systematic review or meta-analysis. While we agree that systematic reviews and meta-analyses 

are valuable tools for synthesizing research, we chose to conduct a scoping review for several reasons. One reason is 

that the current literature on empathy training in healthcare professional education is still relatively new and rapidly 

evolving, and there is not enough information available to conduct a meta-analysis.

Additionally, the methods used in the included studies were often too different to allow for direct comparisons, which is a prerequisite for meta-analysis. A scoping review, on the other hand, allowed us to provide a comprehensive overview of the current state of empathy training in healthcare professional education, including the different methods used and the limitations of the available research. We believe that our scoping review provides an important foundation for future research on empathy training in healthcare professional education and highlights the need for more standardized approaches to training healthcare professionals in empathetic care. While we appreciate the suggestion to conduct a systematic review or meta-analysis, we believe that the approach we took was appropriate given the current state of the literature.

Round 2

Reviewer 1 Report

Dear Authors,
Despite some potential in the collection of papers selected for this review, many changes that could have improved the quality of the article were omitted. Therefore, I regret to inform you that the second version of the manuscript, in my opinion, does not meet the expectations of a major revision. As it stands, this text is of too poor quality to be suggested for approval for publication.
Best regards

Author Response

Academic Editor Notes:

  1. The format of table 1 should be in landscape to allow easy readility. We agree the table is more easily readable in landscape
    1. We are unable to make modifications beyond what has been originally implemented with the Healthcare journal editors. We would appreciate the Healthcare journal editors assistance in formatting as they have originally applied landscape formatting to the original document.
  2. In Table 2, it is crucial that the duration of training should be included, as it ranges from 1 hour to 1 course
    1. Thank you, the ranges are noted in table 2 under ‘Duration’. Previously we were asked to remove this column, currently we have reinserted this column back into the manuscript. This information relayed in our proposed manuscript is noted as is published in the referenced manuscript. The level of detail offered according to hours or days or weeks or semester is the level of detail provided in the referenced manuscript.  This variation between manuscripts supports the need for a unified approach for reporting academic instruction hours particular to empathy education in health sciences education.
  3. About assessment in Table 3 and line 164 onward, I wonder why the authors used US to compare with other countries though there were 50% of the studies are from the US. An overall comparisons will be desirable.
    1. We have made modifications to our verbiage to ensure USA is not used as a comparator to countries outside the US. We are unable to provide you with the requested information regarding the duration and training duration reported in the individual studies, as well as support. Our resources and expertise are limited to the scope of our own research and analysis, and we do not have access to the specific details of the studies you mentioned.
  4. In the discussion, there is no need to underline the phrases. Perhaps these could be used as sub-headings.
    1. Thank you, the formatting has been changed to subheading format.
  5. Line 224, it is not desirable to use USA to compare countries outside the US.
    1. We have made modifications to our verbiage to ensure USA is not used as a comparator to countries outside the US. We appreciate the reviewers' comments and understand their concern regarding the comparability of our findings across different countries. As we acknowledge in our manuscript, differences in education and practice across countries can impact the results of our study. Therefore, we presented this information to allow for comparisons, but we also recognize that the limited number of studies on this topic and the case-based nature of the results limit the more rigorous comparisons we had hoped to conduct. Nonetheless, we believe that our study provides valuable insights into the current state of training programs for pharmacists in pain management across different countries.

Reviewer 2 Report

Most issues were addressed by the authors.

Author Response

(The authors gave the same response as above.)

Reviewer 3 Report

1. Page 1, line 43, I suggest adding an adjective in front of "patient outcomes" like "improved"

2. Page 1 and 2, lines 44-45.  I appreciate the authors adding a reference to paternalism.  This statement should be referenced.  Also, on page 15, lines 169-170

3. Page 17, line 255, the phrase "training healthcare professional studies" should be revised to read "training healthcare professionals" or something like that.

Author Response

(The authors gave the same response as above.)
